# Technical Note: Deep Learning for Creating Surrogate Models of Precipitation in Earth System Models

Theodore Weber[1], Austin Corotan[1], Brian Hutchinson[1,2], Ben Kravitz[3,4], and Robert Link[5]

[1]Computer Science Department, Western Washington University, Bellingham, WA.
[2]Computing and Analytics Division, Pacific Northwest National Laboratory, Seattle, WA.
[3]Department of Earth and Atmospheric Sciences, Indiana University, Bloomington, IN.
[4]Atmospheric Sciences and Global Change Division, Pacific Northwest National Laboratory, Richland, WA.
[5]Joint Global Change Research Institute, Pacific Northwest National Laboratory, College Park, MD.

**Correspondence:** Brian Hutchinson, Communications Facility 495, Computer Science Department, Western Washington University, 516 High Street, MS9165, Bellingham, WA 98225. (brian.hutchinson@wwu.edu)

**Abstract.** We investigate techniques for using deep neural networks to produce surrogate models for short term climate forecasts. A convolutional neural network is trained on 97 years of monthly precipitation output from the 1pctCO2 run (the $CO_2$ concentration increases by 1% per year) simulated by the CanESM2 Earth System Model. The neural network clearly outperforms a persistence forecast and does not show substantially degraded performance even when the forecast length is extended to
120 months. The model is prone to underpredicting precipitation in areas characterized by intense precipitation events. Scheduled sampling (forcing the model to gradually use its own past predictions rather than ground truth) is essential for avoiding amplification of early forecasting errors. However, the use of scheduled sampling also necessitates preforecasting (generating forecasts prior to the first forecast date) to obtain adequate performance for the first few prediction time steps. We document the training procedures and hyperparameter optimization process for researchers who wish to extend the use of neural networks in
developing surrogate models.

## 1 Introduction

Climate prediction is a cornerstone in numerous scientific investigations and decision making processes (e.g., Stocker et al., 2013; Jay et al., 2018). On the long term (decades to centuries), different possible climate outcomes pose very different hazards,
risks, and societal challenges, such as building and maintaining infrastructure (e.g., Moss et al., 2017). On decadal timescales, predictability of major modes of variability (like the El Niño Southern Oscillation) are important drivers of extreme events, such as flooding and drought (e.g., Yeh et al., 2018). On similar timescales, disappearing Arctic sea ice has been implicated in changes in midlatitude winter storm patterns (Cohen et al., 2014). On shorter timescales (weeks to months), also sometimes called the subseasonal-to-seasonal (S2S) regime, climate forecasts can be critical for agriculture, water resource management, flooding/drought mitigation, and military force mobilization (Robertson et al., 2015).

Improvements in climate predictability in some of these regimes are slow to be realized. Decadal predictability studies have found that predictability skill is greatly influenced by proper initialization of hindcasts to ensure that any modeled changes due to internal variability are in phase with observations (Bellucci et al., 2015). This highlights the importance of climate memory in predictive skill, in that the response to processes can be lagged, and the responses themselves can depend upon the model state. Yuan et al. (2018) found the existence of processes with a relatively high portion of response that can be explained by memory on all timescales, from monthly through multidecadal lengths. As an example, Guemas et al. (2013) found that properly initialized hindcasts were able to predict the global warming slowdown of the early 2000s (Fyfe et al., 2016) up to five years ahead.

The best technique for climate prediction is to run an Earth System Model (ESM), as these models capture the state of the art in our knowledge of climate dynamics. These models, however, are difficult and costly to run, and for many researchers access to ESM output is limited to a handful of runs deposited in public archives. Therefore, there has been a great deal of interest in *surrogate models* that can produce results similar to ESMs, but are more accessible to the broader research community due to ease of use and lower computational expense. These surrogates are advantageous for numerous applications, such as exploring wide ranges of scenarios; such efforts are quite costly in ESMs.

Building surrogates of ESMs can take numerous forms. The most basic is pattern scaling (Santer et al., 1990), involving scaling a time-invariant pattern of change by global mean temperature (e.g., Mitchell, 2003; Lynch et al., 2017). Other methods include statistical emulation based on a set of precomputed ESM runs (Castruccio et al., 2014), linear time-invariant approaches (MacMartin and Kravitz, 2016), or dimension reduction via empirical orthogonal functions (e.g., Herger et al., 2015). While all of these methods have shown some degree of success, they inherently do not incorporate information about the internal model state (Goddard et al., 2013), nor can they capture highly nonlinear behavior.

Various methods of incorporating state information into surrogate models have been studied. Lean and Rind (2009) explored using a linear combination of autoregressive processes with different lag timescales in explaining global mean temperature change. However, such reduced order modeling approaches, which explicitly capture certain physical processes, will invariably have limited structure; this can result in inaccurate predictions when evaluating variables like precipitation, which have fine temporal and spatial features, important nonlinear components in their responses to forcing, and decidedly non-normal distributions of intensity and frequency. Many studies have focused on initializing the internal model state of Earth System Models (or models of similar complexity) to capture low frequency variability; this has been found to add additional skill beyond external forcing alone (Goddard et al., 2013). However, the required computational time to create a decadal prediction ensemble is rather expensive.

Machine learning methods offer the possibility of overcoming these limitations without imposing an insurmountable computational burden. The field of machine learning studies algorithms that learn patterns (e.g., regression or classification) from data. The subfield of deep learning focuses on models, typically neural network variants, that involve multiple layers of non-linear transformation of the input to generate predictions. Although many of the core deep learning techniques were developed decades ago, work in deep learning has recently exploded, achieving state-of-the-art results on a wide range of prediction tasks.

This progress has been fueled by increases in both computing power and available training data. While training deep learning models is computationally intensive, trained models can make accurate predictions quickly (often a fraction of a second).

The use of deep learning in climate science is relatively new. Most of the applications have used convolutional neural networks (see Section 3 for further definitions and details) to detect and track features, including extreme events (tropical cyclones, atmospheric rivers, and weather fronts) (e.g., Liu et al., 2016; Pradhan et al., 2018; Deo et al., 2017; Hong et al., 2017) or cloud types (Miller et al., 2018). Other promising applications of deep learning are to build new parameterizations of multi-scale processes in models (Jiang et al., 2018; Rasp et al., 2018) or to build surrogates of entire model components (Lu and Ricciuto, 2019). More generally, McDermott and Wikle (2018) have explored using deep neural networks in nonlinear dynamical spatio-temporal environmental models, of which climate models are an example, and Ouyang and Lu (2018) applied *echo state networks* to monthly rainfall prediction. These studies have clearly demonstrated the power that deep learning can bring to climate science research and the new insights it can provide. However, to the best of our knowledge, there have been few attempts to assess the ability of deep learning to improve predictability, in particular the ability to incorporate short-term memory. Several deep learning architectures (described below) are particularly well suited for this sort of application.

Here we explore techniques for using deep learning to produce accurate surrogate models for predicting the forced response of precipitation in Earth System Models. Key to this approach is training the model on past precipitation data (as described later, a sliding 5-year window), allowing it to capture relevant information about the state space that may be important for predictability in the presence of important modes of variability. The surrogate models will be trained on climate model output of precipitation under a scenario of increasing $CO_2$ concentration and then used to project precipitation outcomes into a period beyond the training period. That forecast will then be compared to the actual climate model output for the same period to quantify performance. Several model designs will be compared to evaluate the effectiveness of various deep learning techniques in this application. The performance of the deep learning surrogates will be compared to other methods of forecasting, such as persistence and autoregressive processes (described later). A key area of investigation will be the prediction horizon (how far out is the predictive skill of the surrogate model better than naive extrapolation methods) for the chosen window size (how much past information is used to condition the surrogate's predictions).

## 2 Study Description

The dataset used for this study was idealized precipitation output from the CanESM2 Earth System Model (Arora et al., 2011). The atmospheric model has a horizontal resolution of approximately 2.8° with 35 vertical layers, extending up to 1 hPa. The atmosphere is fully coupled to the land surface model CLASS (Arora and Boer, 2011) and an ocean model with approximately 1° horizontal resolution. The model output used corresponds to the 1pctCO2 simulation, in which, starting from the preindustrial era, the carbon dioxide concentration increases by 1% per year for 140 years, to approximately quadruple the original concentration. This idealized simulation was chosen to reduce potential complicating factors resulting from precipitation responses to multiple forcings (carbon dioxide, methane, sulfate aerosols, black carbon aerosols, dust, etc.) that might occur under more

comprehensive scenarios, such as the Representative Concentration Pathways (van Vuuren et al., 2011). For this study, only monthly average precipitation output was used; results for daily average precipitation are the subject of future work.

We divided the model output into three time periods. The *training* set consists of the period 1850–1947, and is used to train the surrogate model. The *development* set (sometimes called the validation set) consists of the period 1948–1968, and is used to evaluate the performance of the trained surrogate model to guide further tuning of the model's hyperparameters (i.e., configurations external to the model that are not estimated during training). The *test* set consists of the period 1969–1989 and is used only in computing the end results, which are reported below in Section 4.

## 3   Deep Learning Methodologies for Improving Predictability

Deep learning is a subfield of machine learning that has achieved widespread success in the past decade in numerous science and technology tasks, including speech recognition (Hinton et al., 2012; Chan et al., 2016), image classification (Krizhevsky et al., 2012; Simonyan and Zisserman, 2014; He et al., 2016), and drug design (Gawehn et al., 2016). Its success is often attributed to a few key characteristics. First, rather than operating on predetermined (by humans) features, deep learning is typically applied to raw inputs (e.g., pixels), and all of the processing of those inputs is handled in several steps (*layers*) of non-linear transformation; the addition of these layers increases the *depth* of the network. This allows the model to learn to extract discriminative, non-linear features for the task at hand. Second, all stages of the deep learning models, including the training objectives (defined by a loss function), are designed to ensure differentiability with respect to model parameters, allowing models to be trained efficiently with stochastic gradient descent techniques. With enough training data and computational power, deep learning models can learn complex, highly non-linear input-output mappings, such as those found in Earth System Models.

### 3.1   Architectures

In this work we consider convolutional neural networks (CNNs; LeCun et al., 1998) to model the spatial precipitation patterns over time. CNNs are able to process data with a known grid-like topology, and have been demonstrated as effective models for understanding image content (Krizhevsky et al., 2012; Karpathy et al., 2014; He et al., 2016). Unlike standard fully-connected neural networks, CNNs employ a convolution operation in place of a general matrix multiplication in at least one of their layers. In a convolutional layer, a $m \times n \times p$ input tensor is convolved with a set of $k$ $i \times j \times p$ kernels to output $k$ feature maps that serve as inputs for the next layer. In this setting, $m$ and $n$ correspond to the width and height of the input, and $p$ corresponds to the depth (i.e., number of channels). Similarly, $i$ and $j$ correspond to the width and height of the kernel, and $p$ corresponds to the depth, which is equal to that of the input tensor. In practice we choose a kernel where $i \ll m$ and $j \ll n$ (e.g., $i = j = 3$). By using a small kernel, we limit the number of parameters required by the model while maintaining the ability to detect small, meaningful features in a large and complex input space. This both reduces memory requirements of the model and improves statistical efficiency (Krizhevsky et al., 2012; Goodfellow et al., 2016). The learning capacity of the model can also be adjusted by varying the width (i.e., number of kernels) and the depth (i.e., number of layers) of the network.

**Table 1.** Description of all of the neural network architectures used in this study, including the number of layers, whether the network is a residual or plain network (plain networks do not have shortcut connections), whether scheduled sampling was used, whether preforecasting was used, and the figures of the paper in which each network is used. In addition, these results are compared with persistence forecasting and an autoregressive model. Table A1 provides information about the hyperparameters used for each configuration, and Table A2 describes the hyperparameter space used in model training.

| Name | # layers | Residual? | Sched. Samp.? | Preforecasting? | Figures |
|---|---|---|---|---|---|
| 18-layer residual | 18 | Residual | Yes | Yes | 1,2,3,4,5,6 |
| 18-layer residual (No Scheduled Sampling) | 18 | Residual | No | Yes | 4,5 |
| 18-layer residual (No Preforecasting) | 18 | Residual | Yes | No | 6 |
| 18-layer plain | 18 | Plain | Yes | Yes | 1,4,5 |
| 5-layer plain | 5 | Plain | Yes | Yes | 1,4,5 |

The spatial structure of each time step of the precipitation field bears a resemblance to the structure of image data, making CNNs a promising candidate model. However, to produce accurate forecasts, the model must also incorporate temporal evolution of the precipitation field. To address long-term and short-term trends we implement a sliding window approach where our input tensor is built using precipitation outcomes from the most recent $K$ time steps as input channels. Our model predicts the global precipitation outcome at next time step. Then this procedure is iterated, using output from the previous model forecast as input into the next one, allowing for arbitrarily long prediction horizons.

We consider adding depth to our network because many studies in image classification have achieved leading results using very deep models (Simonyan and Zisserman, 2014; Szegedy et al., 2015; He et al., 2015). Many of the fields of interest in climate science are effectively images, making image classification an appropriate analogue for our aims. In deep residual networks (He et al., 2016), rather than train each $i$ layer to directly produce a new hidden representation $h_{(i)}$ given the hidden representation $h_{(i-1)}$ it was provided, we instead train the layer to produce a residual map $f_{(i)} = h_{(i)} - h_{(i-1)}$, which is then summed with the $h_{(i-1)}$ to give the output for the layer. This way, each layer explicitly refines the previous one. The residual modeling approach is motivated by the idea that it is easier to optimize a residual mapping than to optimize the original, unreferenced mapping. Architecturally, outputting $f_{(i)} + h_{(i-1)}$ at each layer is accomplished by introducing *shortcut* connections (e.g., He et al., 2016) that skip one or more layers in the network. In our CNN, a shortcut connection spans every few consecutive layers. This identity mapping is then summed with the residual mapping $f_{(i)}$ produced by the series of stacked layers encompassed by the shortcut connection. While developing our model, we also explored using plain CNNs (without shortcut connections). The best performing model is described in the following section. All of the neural network architectures considered in this study are summarized in Table 1.

## 3.2 Implementation

Our residual network implementation follows the work in He et al. (2016). We construct a 18-layer residual network with shortcut connections encompassing every two consecutive convolutional layers (see Figure 1). Each convolutional layer uses

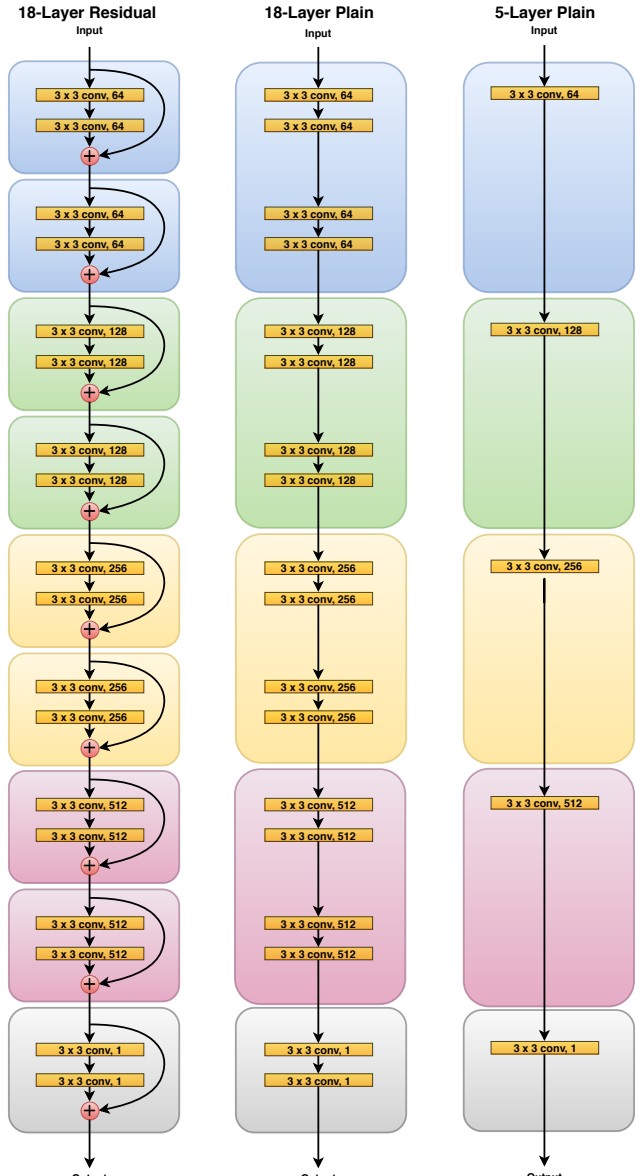

**Figure 1.** Deep architectures for precipitation forecasting. **Left:** 18-layer residual network. **Middle:** 18-layer plain network (no shortcut connections). **Right:** 5-layer plain network (no shortcut connections).

a $3 \times 3$ kernel with a stride of 1, and zero-padding is applied to preserve the spatial dimensionality of the input throughout the network. Directly after the convolution we apply batch normalization following practices in Ioffe and Szegedy (2015), and then ReLU (Nair and Hinton, 2010) as the non-linear activation function. Every four layers we double the number of kernels to increase the learning capacity of our model. When this occurs we zero-pad our identity mapping in the shortcut connection to match dimensions.

The total number of parameters in the 18-layer deep neural network is 34,578, relatively small compared to the size of each input: $60 \cdot 128 \cdot 64 = 491,520$. The length of the training set is 1,116 time steps. For a fixed set of hyperparameters, training each model takes 1-3 hours (depending upon the number of training epochs) on a single NVIDIA 980 ti GPU. The computational cost is effectively all in the training; predictions take a matter of seconds, even on CPU architectures. For comparison, a 5-year simulation in an ESM can take roughly one day even on hundreds of CPUs.

We chose to initialize our network parameters by drawing from a Gaussian distribution as suggested in He et al. (2015). We use stochastic gradient descent to optimize our loss function $L(\theta)$, which is defined as the mean squared area-weighted difference, calculated as

$$L(\theta) = \frac{1}{Z} \sum_x \left[ \left( B(x) - \hat{B}(x;\theta) \right) \cdot A(x) \right]^2 \tag{1}$$

where $x$ iterates over the $Z$ distinct spatial positions, $B(x)$ is the ground truth outcome, $\hat{B}(x;\theta)$ is the CNN reconstruction
(CNN with parameters $\theta$), and $A(x)$ is the cosine of latitude (for area-weighting).

Our plain CNN baselines follow a similar structure as the 18-layer residual network, but all shortcut connections are removed (see Figure 1). By evaluating the performance of the 18-layer plain network, we investigate the benefits of using shortcut connections. We also experiment with a 5-layer plain network to understand how depth affects the performance and trainability of our models. Finally, we train the 18-layer residual network without scheduled sampling (described below) to determine if
this training approach actually improves forecasting ability.

### 3.3    Training

The distribution of our training data (aggregated over the entire $m \cdot n \cdot p$ space) was heavy-tailed and positively skewed (Fisher Kurtosis $\approx 11.3$, Fischer-Pearson coefficient of skewness $\approx 2.72$). Performance of deep architectures tend to improve when training with Gaussian-like input features (Bengio, 2012), so we apply a log transformation on our dataset to reduce skewness.
In addition, we scale our input values between -1 and 1, bringing the mean over the training set closer to 0. Scaling has been shown to balance out the rate at which parameters connected to the inputs nodes learn, and having a mean closer to zero tends to speed up learning by reducing the bias to update parameters in a particular direction (LeCun et al., 2012).

Our model makes fixed-window forecasts: it requires $K$ previous (ground truth or predicted) precipitation outcomes to generate a forecast for the subsequent time step. During training, since ground truth is available for all time steps, one typically
feeds in ground truth data for all $K$ input time steps. After training, when used in inference mode on new data without ground truth, some or all of the $K$ inputs will need to be previous model predictions. Without care, this mismatch can lead to poor extrapolation: the model is not used to consuming its own predictions as input, and any mistakes made in early forecasts will be amplified in later forecasts. Scheduled sampling (Bengio et al., 2015) alleviates this issue by gradually forcing the model to use its own outputs at training time (despite the availability of ground truth). This is realized by a sampling mechanism that randomly chooses to use the ground truth outcome with a probability $\epsilon$, or the model-generated outcome with probability $1 - \epsilon$, when constructing its input. In other words, if $\epsilon = 1$ the model always uses the ground truth outcome from the previous

time step, and when $\epsilon = 0$ the model is trained in the same setting as inference (prediction). As we continue to train the model
we gradually decrease $\epsilon$ from 1 to 0 according to a linear decay function. Practically speaking, this has the effect of explicitly
degrading our initial states at training time; we discuss the implications for our results below. To improve the forecasting
ability of our models, we employed scheduled sampling during training. Scheduled sampling requires a predetermined number
of epochs (to decay properly). For our results that do not use scheduled sampling, we use early stopping (Yao et al., 2007) as a
regularization technique.

    Each model has its own set of hyperparameters with the exception of window size, as modifying window size would result
in each model being conditioned on different numbers of priors, making them difficult to compare fairly. The most significant
hyperparameters in our models are the learning rate, input depth, and number of training epochs. For each model, we tuned
these hyperparameters using Random Search (Bergstra and Bengio, 2012) for 60 iterations each. Our best residual network
used a learning rate of $\sim 0.07$, window size of 60, and was trained for 90 epochs with scheduled sampling. (See Appendix A
for a discussion of window size.) We trained our baseline CNNs using the same window size and similar ranges for the learning
rate and number of training epochs in an attempt to generate comparable models. Each model was trained on a single GPU.

## 4   Predictability and Performance

We evaluate the forecasting ability of our models using the CanESM2 output over the period 1969–1989 as ground truth.
We also compare the performance of our best models against two naive methods of forecasting. The lowest bar, persistence
forecasting, extrapolates the most recent precipitation outcome (i.e., the last outcome in the dev set) assuming the climatology
remains unchanged over time. In perhaps a closer comparison, we also use a first order autoregressive model in each grid box,
abbreviated AR(1), in which the most recent precipitation outcome depends upon a linear combination of the previous value
and a stochastic (white noise) term:

$\hat{B}(x; \theta, i) = c + \phi \hat{B}(x; \theta, i - 1) + \epsilon$                         (2)

where $\hat{B}(x; \theta, i)$ is the surrogate prediction at time $i$, $c$ is a constant, $\epsilon$ represents white noise, and $\phi$ is a parameter that is
tuned/trained such that the model has accurate predictions over the dev set.

    To quantify our generalization error, we compute the root mean square over the area-weighted difference $\hat{B} - B$ for six
different spatial regions — polar, mid-latitude, and tropics over both land and ocean. This is calculated as

$$\text{RMSE} = \sqrt{\frac{\sum_x [(\hat{B}(x; \theta) - B(x)) \cdot A(x) \cdot M_k(x)]^2}{\sum_x [A(x) \cdot M_k(x)]^2}} \tag{3}$$

where $x$ iterates over the spatial positions, $\hat{B}(x; \theta)$ is the surrogate prediction, $B(x)$ is the ground truth outcome, $A(x)$ is the
cosine of latitude weights, and $M_k(x)$ is the region mask (for $k \in \{1, 2, \ldots, 6\}$).

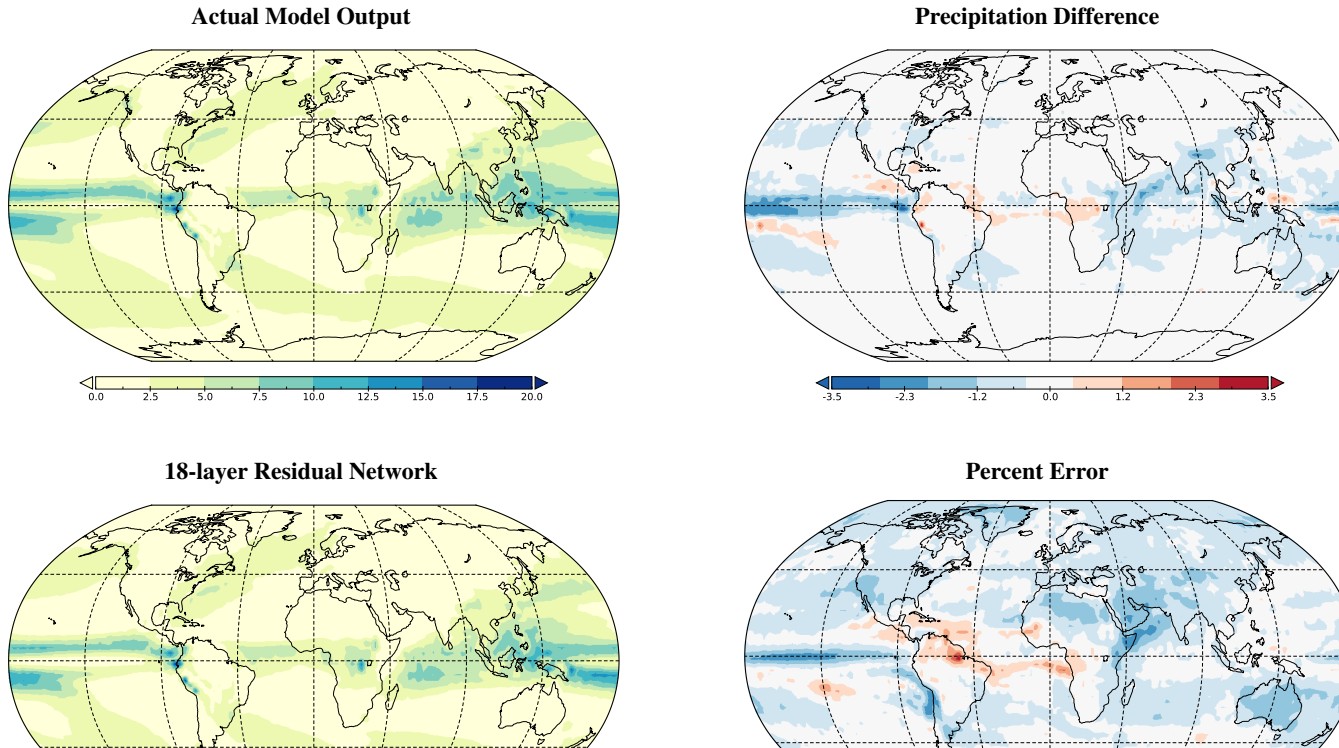

**Figure 2.** Precipitation outcomes (mm day$^{-1}$) for the period 1969–1989. Top shows the average output of the CanESM2 Earth System Model over that period. Bottom shows the average output of a 252-month forecast over the same time period using the 18-layer Residual Network with a window size of 60. Both models show qualitatively similar features.

**Figure 3.** Comparison of the average precipitation outputs (mm day$^{-1}$) of the CanESM2 Earth System ($B$) and the 18-layer Residual Network ($\hat{B}$) using a window size of 60 over the years 1969–1989. Top shows $\hat{B} - B$. Bottom shows the percent error between $\hat{B}$ and $B$. The residual network tends to underpredict near the equator, midlatitude storm tracks, and areas associated with monsoon precipitation.

In addition to RMSE, we also compute the Anomaly Correlation Coefficient (ACC), a commonly used metric for quantifying differences in spatial fields in forecasting (Joliffe and Stephenson, 2003). ACC is defined as (JMA, 2019):

$$\text{ACC} = \frac{\sum_{i=1}^{n}(f_i - \bar{f})(a_i - \bar{a})}{\sqrt{\sum_{i=1}^{n}(f_i - \bar{f})^2 \sum_{i=1}^{n}(a_i - \bar{a})^2}} \tag{4}$$

where $n$ is the number of samples. $f_i$ is the difference between forecast and reference, and $a_i$ is the difference between some verifying value and the reference. We use the average precipitation over the period 1938–1968 as our reference (the 30 years preceding the test set period). $\bar{f}$ and $\bar{a}$ indicate area-weighted averages over the number of samples. ACC can take values in

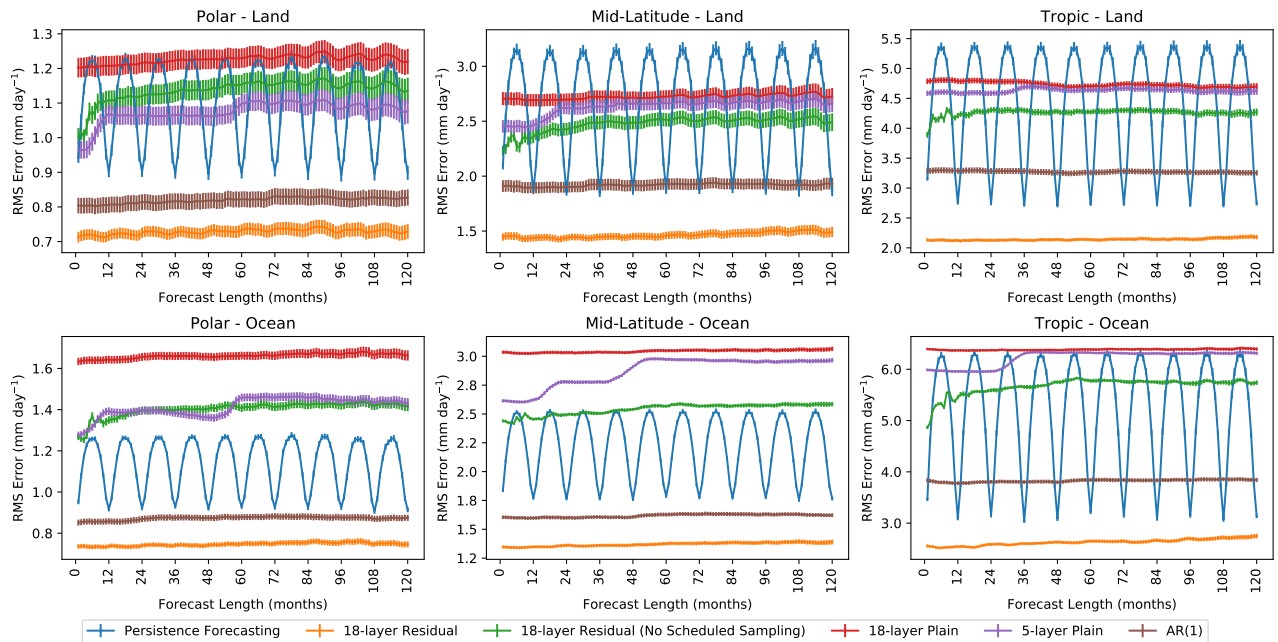

**Figure 4.** RMSE for decadal precipitation forecasts for six regions of interest. Both the plain and residual CNNs used a window size of 60. Vertical bars denote the standard error over all possible starting dates in the test set. The deep residual network with scheduled sampling outperforms all models in all regions and achieves consistent error over time. Removing any of these features from the network results in substantially lower performance.

$[-1, 1]$, where an ACC of 1 indicates that the anomalies of the forecast match the anomalies of the verifying value, and an ACC of -1 indicate a reversal of the variation pattern. Figure 5 shows ACC values for the different models considered in this study. The message is similar to that of Figure 4, with the 18-layer residual network showing the greatest skill (ACC exceeding 0.5 in all six regions), the other neural networks showing little skill, and the persistence forecast showing variable skill, depending upon the forecast length. Although it is difficult to make exact quantitative comparisons, the 18-layer residual network has higher values of ACC than the Community Earth System Model Decadal Prediction Large Ensemble (CESM-DPLE) in all six regions (Yeager et al., 2018). Performance is similar over the Sahel, indicating some ability of the residual network to capture relevant precipitation dynamics.

Figure 2 shows the average precipitation for a 252-month forecast over the period 1969–1989 in the 18-layer Residual Network (CNN with lowest forecasting error) and the average precipitation over the same period in the CanESM2 Earth System Model under a 1pctCO2 simulation. Both models show qualitatively similar features, indicating that the residual network was

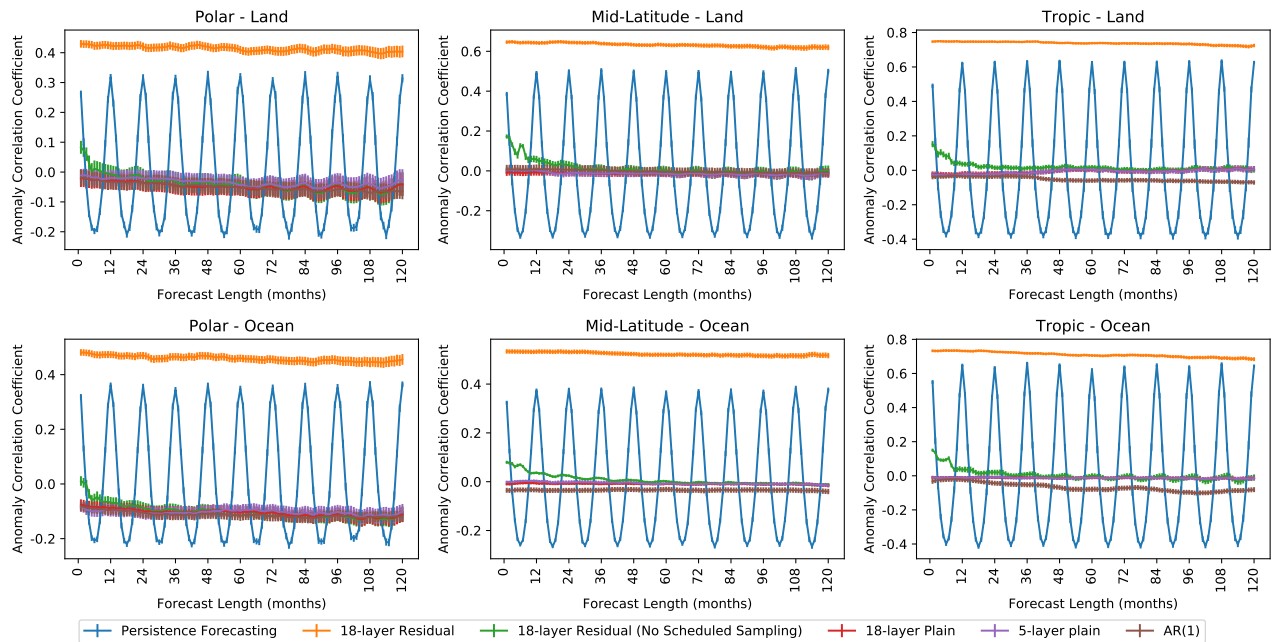

**Figure 5.** Anomaly Correlation Coefficient for decadal precipitation forecasts for six regions of interest. Both the plain and residual CNNs used a window size of 60. Vertical bars denote the standard error over all possible starting dates in the test set. As in Figure 4, the deep residual network with scheduled sampling outperforms all models in all regions, consistently exhibiting a positive correlation with the ground truth outcomes.

capable of reproducing Earth System Model outputs reasonably well. Figure 3 shows the area-weighted difference $\hat{B} - B$ as well as the area-weighted percent error given by

$$pct\_err = \frac{(\hat{B}(x) - B(x)) \cdot A(x)}{B(x) \cdot A(x)} \tag{5}$$

5     The residual model is prone to underpredict near the equator, in the midlatitude storm tracks, and in areas associated with monsoon precipitation. All of these regions experience intense precipitation events (storms), which constitute the right tail of the precipitation distribution. The mean-squared error loss function is less robust to outliers (Friedman et al., 2001, Chapter 10), which are far more common in these regions than others, potentially explaining why the residual network tends to be biased low in these regions. On average, our model achieves reasonably low error on the test set, with a mean precipitation difference

10   of -0.237 mm day$^{-1}$ and mean percent error of -13.22%.

    Figure 4 shows the forecasting performance of each model on a decadal timescale. The 18-layer residual network outperforms all models in all regions, and exhibits relatively consistent error over time. The AR(1) model generally performs second-best in all cases except some seasons in the tropics when persistence tends to perform better. Our plain CNNs have less consistent error over time and performed worse than our residual networks overall. These networks proved more difficult

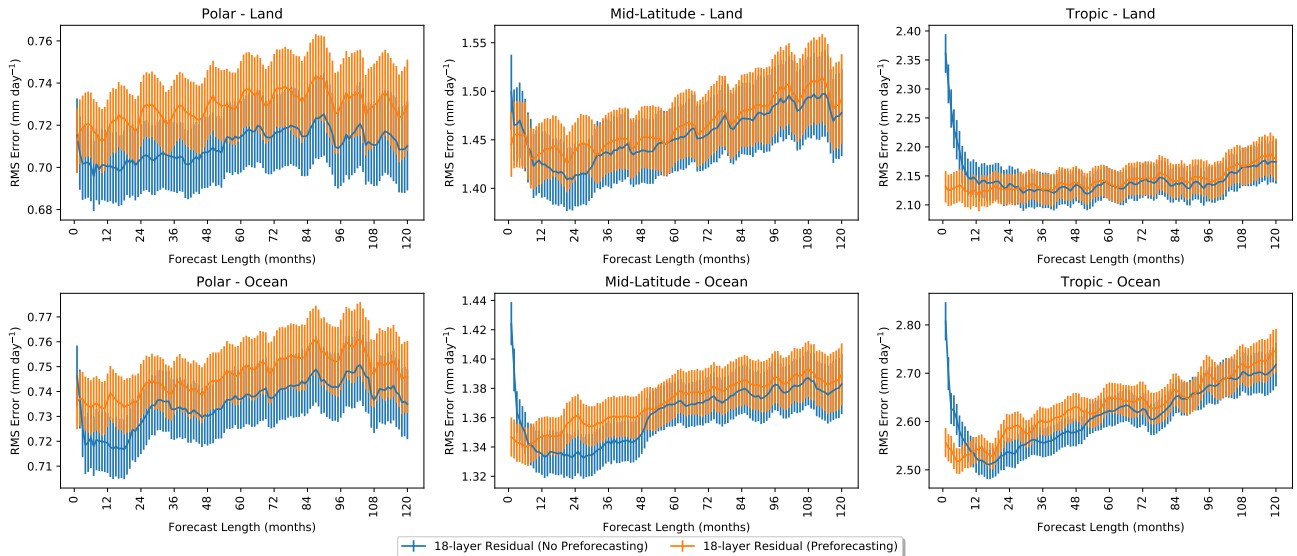

**Figure 6.** Comparing the RMSE for decadal precipitation forecasts with and without preforecasting. The model was trained using a window size of 60, and 30 preforecasts were generated. Vertical bars denote the standard error over all possible starting dates in the test set. Preforecasting significantly reduces the RMSE for short range forecasts without degrading long range forecasts.

to optimize and would often learn to predict similar values at each pixel regardless of spatial location. The 5-layer network showed lower generalization error than the 18-layer network, which was expected behavior as plain networks become more difficult to train with increased depth (He et al., 2016). This challenge is well addressed in our 18-layer residual network, however, as it achieves good accuracy with significant depth.

To assess the benefits of scheduled sampling, we evaluated the performance of an identical residual network architecture trained without scheduled sampling (see "18-layer Residual (No Scheduled Sampling)" in Figure 4). For this model we observe the RMSE quickly increasing during the first few forecasts, indicating that it is not accustomed to making forecasts conditioned on past predictions. Surprisingly, this model also had significantly higher RMSE for the 1-month forecast, which is entirely conditioned on ground truth outcomes. We would expect a model trained without scheduled sampling to perform well in

this case, as the input contains no model generated data. However, as there are sufficient differences in the training setting (i.e., the use of early stopping), it is likely that these models converged to disparate minima. We hypothesize that additional hyperparameter tuning could decrease the RMSE for 1-month forecasts in these models.

Importantly, for the model using scheduled sampling, the skill of the forecast does not change appreciably with lead time, whereas one might expect the model to have some initial-value predictability (e.g., Branstator et al., 2012), and thus more

skillful predictability in the near term. In early forecasts, the input tensors primarily consist of ground truth precipitation outcomes. A model trained using scheduled sampling performs worse in this setting because it is accustomed to seeing inputs that contain model generated data, that is, the initial states are explicitly degraded. This was a choice in terms of the problem we attempted to solve (reducing initial-value predictability in favor of longer forecasts), and scheduled sampling may not be

an appropriate choice for other applications. To address this poor early forecast skill, we explored *preforecasting*—generating
forecasts preceding the first forecast date to prime the model for inference (prediction). By taking this approach we ensure
that the input tensor for the first forecast will contain at least some portion of model generated data. To employ preforecasting,
the number of preceding outcomes generated must be in the range $[1, \ldots, window\_size]$, and should be chosen relative to the
sampling decay function used during training. We suggest generating $window\_size/2$ preforecasts for a model trained using a
linear decay function. We take this approach in Figure 6 and find that it adequately reduces the RMSE for early forecasts while
still maintaining low error for longer forecasts.

## 5 Discussion and Conclusions

This study explored the application of deep learning techniques to create surrogate models for precipitation fields in Earth
System Models under $CO_2$ forcing. From our experiments we found that a CNN architecture was effective for modeling
spatio-temporal precipitation patterns. We also observed increased accuracy with deeper networks, which could be adequately
trained using a residual learning approach. Finally, we found that scheduled sampling (supplemented with preforecasting)
significantly improved long-term forecasting ability, improving upon the commonly used autoregressive model (although we
admit that we could have more thoroughly explored the span of different linear methods, such as higher order AR or ARIMA
models).

It might be expected that the forecast model skill should asymptotically approach persistence as the predictions move farther
from the initial state. We argue three reasons for why our neural network continues to have good skill/low error:

1. Scheduled sampling helps the model extrapolate from its own predictions, reducing errors in later forecasts.

2. Because the model is being trained on numerous time periods in the 1pctCO2 experiment, it is learning some inherent
   properties of precipitation response to $CO_2$ forcing.

3. We are conditioning each prediction on five years worth of data, so it is likely easier for our model to retain signals
   coming from the initial conditions.

Appendix A provides a comparison between using window sizes of 6 and 60 months, with the former showing steadily de-
creasing predictive skill due to its inability to learn the forced response. This is another point of verification for a conclusion
well known in the decadal predictability community: although the initial state is important for decadal predictions, in forced
runs, a great deal of skill is due to the underlying climate signals Boer et al. (2019).

Based on these results we can identify several ways to enhance our current surrogate models, as well as a few promising
deep learning architectures applicable to this work, with the overall goal of understanding which deep learning techniques may
work well for creating surrogates of climate models.

Bengio et al. (2015) proposed three scheduled sampling decay functions: linear, exponential, and inverse sigmoid. Determin-
ing the optimal decay schedule for our problem could have significant effects on model predictability. Weight initialization has
also been proven to affect model convergence and gradient propagation (He et al., 2015; Glorot and Bengio, 2010); therefore

this must be investigated further. Window size was a fixed hyperparameter during tuning, but we cannot rule out its potential impact on forecasting (see Appendix A). Tuning these existing hyperparameters would be the first step in improving results.

5    Incorporating additional features, such as data from Earth System Models with different forcings, global mean temperature, and daily average precipitation, would provide more relevant information and likely improve predictability. These could be incorporated by modifying our input tensor to include these as additional channels. Such augmentations would be an important step toward designing practical, effective surrogate models in the future.

Two architectural features that we may consider adding are dropout and replay learning. Srivastava et al. (2014) showed that 10   adding dropout with convolutional layers may lead to a performance increase and prevent overfitting. Replay learning is widely used in deep reinforcement learning and was shown to be successful in challenging domains (Zhang and Sutton, 2017). We believe that we can apply a similar concept to our architecture, where we train our network on random past input-output pairs so it can "remember" what it learned previously. This technique could aid in alleviating any bias from more recent training data and therefore boost performance.

15   Convolutional Long Short-Term Memory Networks (LSTMs) have had great success in video prediction (Finn et al., 2016) and even precipitation nowcasting (Shi et al., 2015). They offer an alternate strategy for modeling both the spatial and temporal aspects present in our dataset: whereas our model treats the time dimension as channels, and acts upon a fixed length input window, Convolutional LSTMs use recurrent connections to consume the input one time step at a time with no fixed length restriction on the input. This increases the effective depth between information contained in the distant past and the prediction, 20   which may prove beneficial or harmful for our task; we leave it to future work to evaluate their suitability. One could also draw inspiration from the spatial downsampling, upsampling and specific skip connectivity techniques often used work in semantic segmentation (e.g., U-Nets, introduced for biomedical image segmentation; Ronneberger et al., 2015).

Generative Adversarial Networks (GANs) (Goodfellow et al., 2014) have proven to offer impressive generative modeling of grid-structured data. GANs are commonly used with image data for tasks such as super-resolution, image-to-image translation, 25   image generation and representation learning. The effectiveness of GANs to generate realistic data and ability to be conditioned on other variables (Goodfellow, 2016) make them quite appealing for spatio-temporal forecasting.

The results presented here show significant potential for deep learning to improve climate predictability. Applications of this work extend beyond providing a CNN forecasting library, as there are several enhancements that could yield a more practical alternative to traditional Earth System Models. Ability to emulate alternate climate scenarios, for example, is desirable. Regardless, using an approach that can incorporate the internal model state appears to have promise in increasing prediction accuracy.

*Code and data availability.*  All models were developed in Python using the machine learning framework TensorFlow, developed by Google. 5   Training and inference scripts are available at https://github.com/hutchresearch/deep_climate_emulator. Code used to generate the figures in this paper is available upon request. All climate model output used in this study is available through the Earth System Grid Federation.

**Table A1.** Hyperparameters for each network architecture used in this study (See Table 1). For each architecture, we tuned the learning rate, standard deviation used for weight initialization (sampling from a truncated normal distribution), and number of training epochs (unless training without scheduled sampling, in which case we used early stopping with a patience threshold = 10). For each architecture, the optimal hyperparameter configuration was selected after 60 iterations of Random Search.

| Architecture | Decay Function | Window Size | Learning Rate | Standard Deviation (for Xavier Initialization) | Epochs |
|---|---|---|---|---|---|
| 18-layer residual | Linear | 60 | 0.069 | 0.016 | 90 |
| 18-layer residual (No Scheduled Sampling) | N/A | 60 | 0.095 | 0.021 | 82 (early stopping) |
| 18-layer plain | Linear | 60 | 0.013 | 0.01 | 100 |
| 5-layer plain | Linear | 60 | 0.049 | 0.01 | 125 |

## Appendix A: Effects of Window Size

We briefly explored the effects of different window sizes on predictability for the 18-layer residual network (without scheduled sampling). Figure A1 shows a comparison between a window size of 6 months versus the 60 month window that we used in our best performing model. With a smaller window size, the forecast model is unable to learn enough of the forced response to improve predictive skill, so the forecasts model skill approaches that of persistence.

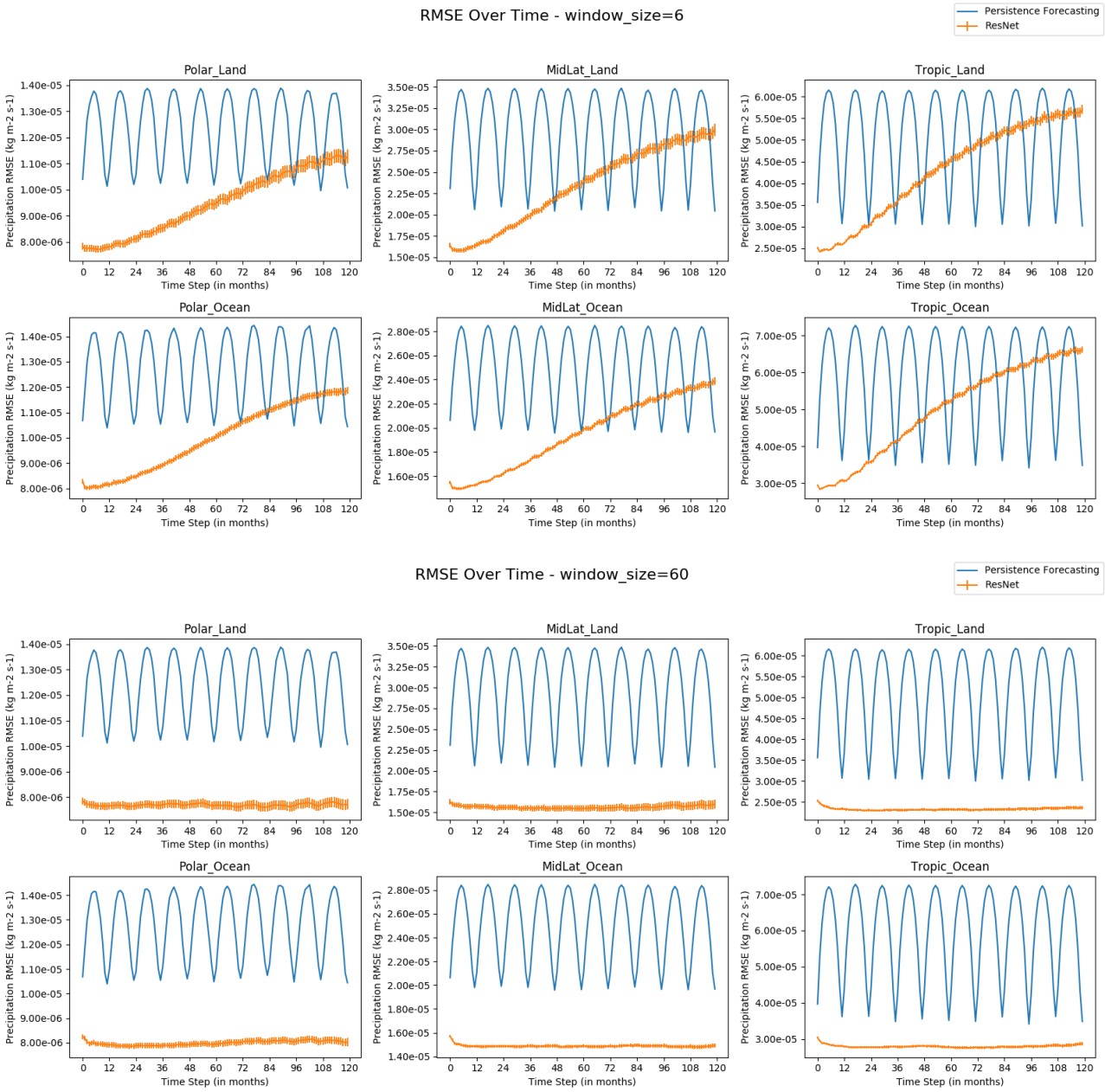

**Figure A1.** Comparing the RMSE for decadal precipitation forecasts in the 18-layer residual network with a 6-month window (top) and a 60-month window (bottom). Both networks were trained using scheduled sampling, and preforecasting was not employed when generating predictions.

**Table A2.** Hyperparameter space used for training the models described in Table 1. For more information on how the hyperparameter space is defined, see https://github.com/hyperopt/hyperopt/wiki/FMin#21-parameter-expressions.

| Hyperparameter | Min | Max | Step | Sampled from | Comment |
|---|---|---|---|---|---|
| Epochs | 75 | 150 | 5 | Quantized Uniform | |
| Learning Rate | 0.001 | 0.01 | N/A | Uniform | |
| Standard Deviation | 0.001 | 0.1 | N/A | Uniform | |
| Window Size | 6 | 120 | N/A | Choice | Only varied during initial experiments; fixed at 60 for the study |

*Competing interests.* None.

*Acknowledgements.* We thank Nathan Urban and an anonymous reviewer for helpful comments. The research described in this paper was supported in part under the Laboratory Directed Research and Development Program at Pacific Northwest National Laboratory, a multiprogram national laboratory operated by Battelle for the U.S. Department of Energy. The Pacific Northwest National Laboratory is operated for the U.S. Department of Energy by Battelle Memorial Institute under contract DE-AC05-76RL01830. Support for B.K. was provided in part by the National Science Foundation through agreement CBET-1931641, the Indiana University Environmental Resilience Institute, and the *Prepared for Environmental Change* Grand Challenge initiative. Finally, the authors would like to thank Md. Monsur Hossain and Vincent Nguyen from Western Washington University for their contributions to model development, and the Nvidia Corporation for donating GPUs used in this research.

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
