# Peer review of "Technical Note: Deep Learning for Creating Surrogate Models of Precipitation in Earth System Models"

_Atmospheric Chemistry and Physics, 2019_

## Referee Comment (RC1) · Nathan Urban (Referee) · 9 May 2019

This is an interesting paper on the under-studied field of deep learning for climate forecasting. I am having some trouble evaluating the skill of the predictions, and have concerns about how the skill changes (or rather, doesn't) with lead time.

Major comments:

I am not sure the best skill comparisons are being made. Persistence forecasting is a low bar to meet. It would be better to compare to some time series forecasting method that has memory, like a damped persistence forecast from an AR(1) model fit to each

grid cell, or more generally, a multi-lag ARIMA model or something with automatic model order selection, or Holt-Winters exponential smoothing. (I acknowledge that this isn't always done in other papers, but I feel it's a stronger comparison, and not necessarily that hard to do.)

Also, the data don't appear to be deseasonalized, which leads to odd-looking oscillations in comparisons to persistence, making it harder to compare.

I would expect the forecast model skill to asymptotically approach persistence as dependence on the initial state is lost, but this doesn't appear to happen. Why not? Is it an artifact of the way persistence is being defined? Is it because there is some skill in the non-stationary forced response (these are transient runs, not control) that the neural network learns? In general, this particular point requires more careful treatment, since a lot of skill at decadal timescales can be due to the climate signal, not actual initial-state predictability. I would expect this to be less true for precipitation, which doesn't exhibit very strong decadal trends, but the point should still be discussed. It may have been better to use an unforced control run as training data, unless the point of the paper is to evaluate predictability coming from the forced response.

Even more worryingly, the skill of the forecast doesn't seem to really change with lead time, for the 18-layer ResNet. Shouldn't it be more skillful in the near term? Is there any initial-value predictability here at all? If not, where is the supposed skill improvement over persistence coming from? This continues to make me concerned that there is something problematic with how skill is being defined relative to persistence.

Other comments:

There should be more reference to the decadal prediction literature. Yeager et al., "Predicting near term changes ...", BAMS (2018) is one recent paper on initialized numerical model forecasts. They give useful diagnostics in addition to RMS error, like anomaly correlation coefficients. This manuscript should compare the predictability found by the neural network method to the predictability found by other decadal forecasts of precipitation, as in Yeager et al. (e.g. Fig. 5), or other papers — this is just one of many.

There are also prior examples of using neural networks (such as recurrent networks) for climate forecasting, for example McDermott and Wikle, Enivronmentrics (2018); Ouyang and Lu, Water Resources Management (2018).

I was confused by some of the exposition. It took a long time to understand that the forecasts are being made on the basis of 5 years (60 months) of precipitation data ... some details of the setup are introduced too late.

The scheduled sampling exposition was particularly confusing. At first I didn't understand why you wouldn't have all of the ground truth data available for forecasts. Then I realized that a fixed-window forecast (as opposed to a recurrent forecast) wouldn't have the full 60 months of data available; it would be using some model forecasts. This should be made more explicit, and also clarify a few details like what "ground truth" data is here (perfect-model data at lags before the forecast). Also, it would help to point out that "inference" is what climate scientists call "prediction". (To statisticians, "inference" often means "training" to a computer scientist.)

I'm not sure what Figure 2 is showing. The average of many 1-month-ahead forecasts take over 21 years of initial conditions? I don't know if this is a useful comparison. I would expect the average of bunch of very short-term forecasts will resemble the average of the actual data. Perhaps I am misunderstanding.

How many neural network parameters are there in total using this architecture, compared to the size of the data? Are there generalization error plots to assess overfitting?

---

## Referee Comment (RC2) · Anonymous Referee #1 · 28 May 2019

General comments: 1. I am not sure how meaningful this study is. This study used a sliding window approach to predict global precipitation, i.e., using CNN to simulate the relationship between the precipitations from the most recent K time steps and that at next time step. First, the mapping becomes useless when we need to predict more than one step into the future or to use more/less than K previous time steps. Secondly, the sliding window approach used the fixed window size, incapable of learning the temporal dependence in a dynamic form. 2. I found the description of methodology and numerical experiments is confusing. After reading the manuscript, I am not sure how many network architectures and how many numerical experiments the authors considered. A table listing all of this information would be very helpful. 3. The comparison with persistence forecasting is not enough to demonstrate the effectiveness and advantages of the deep neural networks. I think a comparison with other advanced time series forecasting methods is necessary, such as autoregression, moving average, and their combinations, and even the more advanced long short-term memory. 4. What is the computational cost to build the surrogate model, such as the number of training samples, the training time, the hyperparameter tuning time? When comparing the methods, besides accuracy, computational costs should be another factor to be considered.

Specific comments: 1. Page 5, Line 1, whether a deep network is needed depends on the problem, i.e., adding depth to the network can improve the model performance of this study. The reason should not be that deep models were successful in recent studies in image classification. As problems are different and the training data size is different, the deep network might not be a good choice of this work. I would like to see a better justification for using the deep network in this work. 2. Page 6, Line 1, If I understand correctly, the training data are 3D images with size m*n*p. What do the authors mean by saying that "The distribution of training data was heavy-tailed and positively skewed"? 3. Page 7, Lines 8-11, I do not understand why not using the ground truth all the time, as errors made in early forecasts would accumulate in later forecasts if the predicted values are used. 4. Page 7, lines 18-19, the comparison is not fair because the baseline CNNs used the best hyperparameters of the residual network. The best set of hyperparameters tuned for the residual network could be a bad choice for the baseline CNNs.

---

## Author Comment (AC1) · 19 Nov 2019

Please see supplement.

Please also note the supplement to this comment:
https://www.atmos-chem-phys-discuss.net/acp-2019-85/acp-2019-85-AC1-supplement.pdf
* * *

---

## Author Comment (AC2) · 19 Nov 2019

Response to reviewers
acp-2019-85
Weber et al.

Original reviewer comments in normal typeface.  **Responses in bold.**
* * *
Reviewer #1 (Nathan Urban)

This is an interesting paper on the under-studied field of deep learning for climate forecasting. I am having some trouble evaluating the skill of the predictions, and have concerns about how the skill changes (or rather, doesn't) with lead time.

**Thanks!  We address the reviewer's comments below.**

Major comments:
I am not sure the best skill comparisons are being made. Persistence forecasting is a low bar to meet. It would be better to compare to some time series forecasting method that has memory, like a damped persistence forecast from an AR(1) model fit to each grid cell, or more generally, a multi-lag ARIMA model or something with automatic model order selection, or Holt-Winters exponential smoothing. (I acknowledge that this isn't always done in other papers, but I feel it's a stronger comparison, and not necessarily that hard to do.)

**We agree with this comment and have added an AR(1) model fit to each grid cell, as the reviewer suggests, as well as a short description.  Figure 4 now shows this comparison. We agree that doing this was valuable, and it does not change our conclusions.**

Also, the data don't appear to be deseasonalized, which leads to odd-looking oscillations in comparisons to persistence, making it harder to compare.

**We acknowledge the reviewer's point.  We choose not to deseasonalize the neural network training data, as we wanted to check whether, with sufficient data and depth in the network, we could capture seasonality.  We agree that this makes odd oscillations in the persistence forecasts, making comparison difficult.  Adding the AR(1) model seems to improve our ability to make comparisons with more "traditional" methods of time series forecasting.**

I would expect the forecast model skill to asymptotically approach persistence as dependence on the initial state is lost, but this doesn't appear to happen. Why not? Is it an artifact of the way persistence is being defined? Is it because there is some skill in the non-stationary forced response (these are transient runs, not control) that the neural network learns? In general, this particular point requires more careful treatment,

since a lot of skill at decadal timescales can be due to the climate signal, not actual initial-state predictability. I would expect this to be less true for precipitation, which doesn't exhibit very strong decadal trends, but the point should still be discussed. It may have been better to use an unforced control run as training data, unless the point of the paper is to evaluate predictability coming from the forced response.

**We acknowledge the reviewer's point, both in terms of predictability and in purpose of the manuscript. We first point out that the point of the paper is to evaluate predictability from the forced response. We have gone back through the manuscript to ensure that this message is coming across.**

**Regarding performance, we had similar concerns when first analyzing the results from our highest performing model. There are a few reasons why we are seeing this behavior:**
**1) Scheduled sampling helps the model extrapolate on its own predictions, reducing errors in later forecasts.**
**2) The model is learning some inherent properties of the precipitation response to forced $CO_2$ increase, as it is trained on data at various time periods in the 1pctCO2 experiment.**
**3) We condition each prediction on 5 years' worth of data, so it may be easier for our model to retain signals coming from the initial conditions.**

**We also experimented with smaller window sizes (6-12 months). For those models, even with scheduled sampling, the forecasting skill did approach persistence:**

[Figure]

**We agree with the reviewer that we could have explained all of this better and have modified the manuscript to better describe these issues.**

Even more worryingly, the skill of the forecast doesn't seem to really change with lead time, for the 18-layer ResNet. Shouldn't it be more skillful in the near term? Is there any initial-value predictability here at all? If not, where is the supposed skill improvement

over persistence coming from? This continues to make me concerned that there is something problematic with how skill is being defined relative to persistence.

**We think this comment stems from insufficient explanation on our part. By using scheduled sampling during training we are forcing the model to become less dependent on the initial-state, as we are explicitly degrading our initial states at training time (by incorporating model predictions in our input tensors). This was a choice on our part based on the problem we aimed to solve, and it also explains why our model doesn't perform better in the near term. We also point out that our model does produce reasonable precipitation outcomes with low error (by our metrics), so clearly our model is able to learn some underlying features of the forced scenario. We have gone back through the manuscript to make sure these points are coming across sufficiently well.**

Other comments:
There should be more reference to the decadal prediction literature. Yeager et al., "Predicting near term changes ...", BAMS (2018) is one recent paper on initialized numerical model forecasts. They give useful diagnostics in addition to RMS error, like anomaly correlation coefficients. This manuscript should compare the predictability found by the neural network method to the predictability found by other decadal forecasts of precipitation, as in Yeager et al. (e.g. Fig. 5), or other papers — this is just one of many.

**We agree with the reviewer's point. We have added measures of ACC to the manuscript, and we also discuss our results on predictability in the context of Yeager et al., which we agree provides a nice overview of decadal forecasts of precipitation.**

There are also prior examples of using neural networks (such as recurrent networks) for climate forecasting, for example McDermott and Wikle, Environmetrics (2018); Ouyang and Lu, Water Resources Management (2018).

**We thank the reviewer for pointing these out and have added citations to them.**

I was confused by some of the exposition. It took a long time to understand that the forecasts are being made on the basis of 5 years (60 months) of precipitation data ... some details of the setup are introduced too late.

**We appreciate this point and have added a mention of this earlier in the manuscript.**

The scheduled sampling exposition was particularly confusing. At first I didn't understand why you wouldn't have all of the ground truth data available for forecasts. Then I realized that a fixed-window forecast (as opposed to a recurrent forecast) wouldn't have the full 60 months of data available; it would be using some model forecasts. This should be made more explicit, and also clarify a few details like what "ground truth" data is here (perfect-model data at lags before the forecast). Also, it would help to point out that "inference" is what climate scientists call "prediction". (To statisticians,

"inference" often means "training" to a computer scientist.)

**We have attempted to clarify the description of the scheduled sampling, and we thank the reviewer for this helpful phrasing. We also thank the reviewer for pointing out that the definition of inference may not be clear - we have added clarity to the manuscript where we use this word.**

I'm not sure what Figure 2 is showing. The average of many 1-month-ahead forecasts take over 21 years of initial conditions? I don't know if this is a useful comparison. I would expect the average of bunch of very short-term forecasts will resemble the average of the actual data. Perhaps I am misunderstanding.

**We apologize for the misunderstanding. Figure 2 compares the average precipitation over all months in the test set vs the average precipitation over a 252-month forecast from our model over the same time period. We have clarified this in the manuscript and the caption of the figure.**

How many neural network parameters are there in total using this architecture, compared to the size of the data? Are there generalization error plots to assess overfitting?

**The total number of parameters in the network is 34,578, as compared to the size of each input: 60*128*64=491,520. The length of the training set is 1,116 timestamps. We have now added this to the manuscript.**

**To some degree, Figure 4 addresses the issue of generalization error, as it shows the model performance against the held-off test set. If the reviewer has something more specific in mind, we would be happy to consider it.**
* * *
Reviewer #2

General comments:

1. I am not sure how meaningful this study is. This study used a sliding window approach to predict global precipitation, i.e., using CNN to simulate the relationship between the precipitations from the most recent K time steps and that at next time step. First, the mapping becomes useless when we need to predict more than one step into the future or to use more/less than K previous time steps. Secondly, the sliding window approach used the fixed window size, incapable of learning the temporal dependence in a dynamic form.

**We thank the reviewer for this comment. We should have been more clear - the procedure can then be iterated, using the output from previous model forecasts as**

**inputs into the next one, so that one can predict far beyond just the next time step. We have added more description to this effect to the manuscript.**

**We do not agree with the reviewer's second point. A fixed window size of 60 months does prevent us from learning dynamical behavior on scales longer than 60 months, but behavior on shorter timescales is incorporated into the training process. We are not entirely sure how this confusion arose, and we would appreciate clarification as to where we should improve our description.**

2. I found the description of methodology and numerical experiments is confusing. After reading the manuscript, I am not sure how many network architectures and how many numerical experiments the authors considered. A table listing all of this information would be very helpful.

**We agree with this comment and have added a table as the reviewer suggests. We have also gone through the relevant sections and improved the clarity of our descriptions.**

3. The comparison with persistence forecasting is not enough to demonstrate the effectiveness and advantages of the deep neural networks. I think a comparison with other advanced time series forecasting methods is necessary, such as autoregression, moving average, and their combinations, and even the more advanced long short-term memory.

**We agree with this comment, and a similar issue was raised by Reviewer #1. We have now added a comparison to an autoregressive model. Please also see the response to Reviewer #1 above.**

4. What is the computational cost to build the surrogate model, such as the number of training samples, the training time, the hyperparameter tuning time? When comparing the methods, besides accuracy, computational costs should be another factor to be considered.

**We agree that computational cost is a factor that should be considered. In our experiments the training set contained 1116 observations. Training time is dependent on the number of training epochs (a hyperparameter of the model), but on average takes 1-3 hours per model, depending on the number of epochs (on a single NVIDIA 980 ti GPU). Predictions, on the other hand, take seconds - the bulk of the cost is in the training. We recognize that our residual network may incur a larger computational cost at training time compared to other surrogate models but is justified by an increase in prediction accuracy. We have modified the manuscript to address this point.**

Specific comments:

1. Page 5, Line 1, whether a deep network is needed depends on the problem, i.e., adding depth to the network can improve the model performance of this study. The reason should not be that deep models were successful in recent studies in image classification. As problems are

different and the training data size is different, the deep network might not be a good choice of this work. I would like to see a better justification for using the deep network in this work.

**We agree with the reviewer, and we did not intend to say that climate science and image classification are identical. The climate system is a complex, nonlinear system, which makes it an ideal candidate for deep neural networks. We do believe that image classification is an appropriate analogue for our purposes, as many climate fields are similar to images or movies. We have clarified this point in the manuscript.**

2. Page 6, Line 1, If I understand correctly, the training data are 3D images with size m*n*p. What do the authors mean by saying that "The distribution of training data was heavy-tailed and positively skewed"?

**We have now clarified in the manuscript that the distribution was computed over the entire m*n*p space.**

3. Page 7, Lines 8-11, I do not understand why not using the ground truth all the time, as errors made in early forecasts would accumulate in later forecasts if the predicted values are used.

**We acknowledge that our description of what we mean by "ground truth" was lacking. We have clarified our description of scheduled sampling to better convey how this works and why errors do not accumulate.**

4. Page 7, lines 18-19, the comparison is not fair because the baseline CNNs used the best hyperparameters of the residual network. The best set of hyperparameters tuned for the residual network could be a bad choice for the baseline CNNs.

**We agree with the reviewer's point about fairness. The only hyperparameter that was fixed across the baseline models was the window size. If we modified window size for each model, the models would be conditioned on more (or fewer) priors, making them difficult to compare fairly. We have now clarified this in the manuscript.**

---

## Author Response (AR2)

Response to second round of reviews
Weber et al.

Original reviewer comments in normal typeface. **Responses in bold.**

In my first review, I questioned the significance of this work. After reading the revised manuscript, I think some effort is still needed to emphasize its significance.

**We thank the reviewer for this comment and are happy to make further improvements to the manuscript.**

1. This work aims to use machine learning methods to build a surrogate model of ESM so as to reduce the computational cost. First, I do not understand how their machine learning methods can reduce the ESM simulation expense. The surrogate model is built based on ESM outputs which means that ESM simulations are still needed. Second, I do not see a computational cost comparison between using and not using the surrogates, i.e., how much cost can be saved after building the surrogates?

**We thank the reviewer for pointing out a way in which clarity in the manuscript was lacking. It is true that one needs training data for the emulator, and that training data comes in the form of ESM output. However, once the emulator is trained, it can provide data for time periods or forcings that have not been simulated with ESMs, and it can do so in a fraction of the time. A 5-year simulation (the maximum length of time we projected in our study) in an ESM can take a day to simulate on hundreds of processors using a supercomputer. Once trained, our emulator can produce a 5-year simulation in seconds on a laptop. We did not do a formal computational cost comparison because such comparisons are highly dependent upon the supercomputer architecture, but we will provide such order-of-magnitude estimates in the manuscript, as well as further clarification of our points.**

2. Given the precipitation data is a time series having temporal dependence, I am wondering why the authors did not use the Convolutional LSTM for their work, as they pointed out that convolutional LSTM is a good for their work which can address both the spatial and temporal aspects of the data.

**We appreciate the reviewer's point, and we acknowledge that we did not explain things clearly. We chose our fixed window architecture because we felt the non-recurrent architecture would have better trainability and that our window size would be provide sufficiently large context for prediction. Because Convolutional LSTMs feed in inputs one step at a time, they also increase the effective depth separating the prediction and distant past timesteps, which can either benefit or harm performance. Now that we have achieved success with our network, we feel confident that taking on an alternative spatio-temporal model would be an interesting addition. We could have said this better in the paper and now do so.**

[revised manuscript text omitted]